



# High-resolution satellite-based cloud detection for the analysis of land surface effects on boundary layer clouds

Julia Fuchs[1,2], Hendrik Andersen[1,2], Jan Cermak[1,2], Eva Pauli[1,2], and Rob Roebeling[3]

[1]Karlsruhe Institute of Technology (KIT), Institute of Meteorology and Climate Research, Karlsruhe, Germany
[2]Karlsruhe Institute of Technology (KIT), Institute of Photogrammetry and Remote Sensing, Karlsruhe, Germany
[3]European Organisation for the Exploitation of Meteorological Satellites (EUMETSAT), Darmstadt, Germany

**Correspondence:** Julia Fuchs (julia.fuchs@kit.edu)

**Abstract.** Continental boundary layer clouds play an essential role in the climate system and are driven by processes linked to the land surface. The observation of boundary layer clouds with high-resolution satellite data can provide comprehensive insights into spatiotemporal patterns of land surface-driven modification of cloud occurrence, such as the diurnal variation of the occurrence of fog holes and cloud enhancements attributed to the impact of the urban heat island. High-resolution satellite-

5 based cloud masking approaches are often based on locally-optimized thresholds that are compared against satellite-observed visible and/or infrared radiances to separate cloudy from clear-sky observations that can be affected by the local surface reflectance. Therefore, spatial differences in surface albedo, as found in and around urban areas or forests, can introduce spatial biases in the detected cloud cover that may impede the analysis of spatial pattern changes due to land surface influences. In this study, two approaches for cloud masking using the High Resolution Visible channel of the Spinning Enhanced Visible and

10 Infrared Imager aboard Meteosat Second Generation are developed and validated for the region of Paris to show and improve applicability for analyses of urban effects on clouds. Firstly, the Local Empirical Cloud Detection Approach (LECDA) uses an optimized threshold to separate the distribution of visible reflectances into cloudy and clear sky for each individual pixel accounting for its locally specific brightness. Secondly, the Regional Empirical Cloud Detection Approach (RECDA) uses visible reflectance thresholds that are independent of surface reflection at the observed location. Validation against in-situ

cloud fractions reveals that both approaches perform similarly with a Heidke Skill Score of 0.69 and 0.71, respectively. While the LECDA is representative for the widespread usage of locally-optimized approaches, comparison against RECDA reveals that the cloud masks obtained from LECDA can result in regional biases of +-5 % that are caused by the differences in surface reflectance. This makes the regional approach RECDA a more appropriate choice for the high-resolution satellite-based analysis of cloud cover changes over different surface types and the interpretation of locally induced cloud processes.



# 1 Introduction

The detection of continental boundary-layer clouds based on satellite data has a long tradition and is essential for the analysis of the various ways of how clouds interact with the land surface from the climate to the micro scale. Different land cover types, such as vegetation and urban surfaces exhibit distinct physical surface properties that influence the latent and sensible heat fluxes between the surface and the boundary layer and thus cloud development and spatial patterns (Shepherd, 2005; Collier, 2006; Varentsov et al., 2018).

Frequently, satellite-based analyses of the impact of land surface characteristics on boundary-layer clouds are based on comparisons of cloud fraction and its spatial anomalies with respect to different land cover types (Teuling et al., 2017; Theeuwes et al., 2019; Pauli et al.). Characteristic modifications of spatial patterns of cloudiness can emerge among others over urban areas. For example, an afternoon-cloud cover enhancement has been found over Paris during summer (Theeuwes et al., 2019), whereas fog frequencies have been observed to be lower over the urban areas of the Gangetic Plain during winter (Gautam and Singh, 2018). In this context, the urban heat island effect has been found to be a main driver of the boundary-layer cloud enhancement over Paris (Theeuwes et al., 2019). In many cases, the mechanisms by which the land-surface characteristics interact with the spatial patterns in cloudiness are not well understood (Zhong et al., 2017; Fan et al., 2016; Liang et al., 2018; Shepherd, 2005; Collier, 2006; Han et al., 2014).

Satellite data are ideally suited to map patterns of cloudiness and facilitate the analysis of land surface-atmosphere interactions at various scales. The High Resolution Visible (HRV) channel of the Spinning Enhanced Visible and Infrared Imager (SEVIRI) sensor onboard Meteosat Second Generation (MSG) is particularly useful in this context due to its high temporal (∼15min) and spatial resolutions (1km at nadir) (Klüser et al., 2008; Deneke and Roebeling, 2010; Derrien et al., 2010; Henken et al., 2011). High resolution cloud masks enable the detection and analyses of modifications of spatial cloud patterns induced by small-scale (smaller than ∼3 km) features. While the higher spatial resolution is a clear advantage of the HRV channel, cloud detection from the single-spectral HRV channel is hampered by missing multi-spectral information that is available at coarser resolutions (e.g. 3 km at nadir for SEVIRI). This is shown by a large number of cloud detection algorithms that are based on multi-spectral thresholding tests to separate clear sky from different cloud types (Ackerman et al., 1998; Rossow and Garder, 1993; Saunders and Kriebel, 1988; Stowe et al., 1999; Di Vittorio and Emery, 2002; Chen et al., 2003; Hutchison et al., 2005; Cermak, 2006; Yang et al., 2007; Bley and Deneke, 2013; Andersen and Cermak, 2018).

HRV-based cloud detection methods to derive cloud masks are rare and challenging due to the limitation of observations from a single visible channel only (Schulz et al., 2012; Nilo et al., 2018). Cloud masks obtained from a single channel are often histogram-based, and seek to separate the clear sky and cloudy components of the reflectivity distribution. The separation of the distribution is typically based on a threshold that is determined by a histogram-based minima approach in a representative data set of a specific surface type and geographic region (Minnis and Harrison, 1984; Ipe et al., 2003; Bley and Deneke, 2013; Cermak, 2006; Cermak and Bendix, 2008). For the empirical determination of the clear sky reflectivity the frequent occurrence of atmospheric and non-atmospheric features can be challenging, as aerosols, cloud shadows, thin cirrus clouds as well as fresh snow affect the retrieved reflectances (Yang et al., 2007; Ipe et al., 2003; Matthews and Rossow, 1987). In general, thresholds





determined by such techniques are temporally dynamic, to account for the diurnal and seasonal variations of the solar zenith angle (Yang et al., 2007) and the vegetation period (Teuling et al., 2017; Theeuwes et al., 2019). A main problem for cloud detection is the influence of the land surface signal on the clear sky determination that can lead to severe misinterpretation of cloud occurrence over different surface types. For example, over bright surfaces a reduced contrast between clear sky reflection and cloudy-sky reflection can lead to an underdetection of clouds (EUMETSAT, 2019). To account for location-specific

differences of surface albedo and to reduce the local bias, thresholds determined over land are frequently locally optimized. Teuling et al. (2017) suggested an empirical approach that flags an observation as cloudy when reflectivity exceeds a locally (per pixel) and temporally (per hour, 10-day period) adjusted clear-sky climatological value. Such locally adjusted and empirically based thresholds have a low bias in total but, still, are challenged by a degradation of detection accuracy depending on the local surface brightness (Bley and Deneke, 2013). This is caused by locally-optimized approaches that seek to find the

optimal threshold to distinguish clouds and clear sky and thus have to deal with a surface-dependent bias (see explanation in Sect. 3.2 and 1b)). For analyses of land-surface impacts on clouds it is, however, crucial that the satellite-based detection of cloud occurrence over different land cover types is unbiased with respect to surface properties.

In this study, two approaches to detect boundary-layer clouds on the basis of SEVIRI HRV data are presented and compared for the analysis of cloud cover changes over different surface types: the locally optimized cloud detection scheme of the Lo-

cal Empirical Cloud Detection Approach (LECDA) and the Regional Empirical Cloud Detection Approach (RECDA) that is robust to regional variations in surface reflectivity. Both approaches are processed for November of a multi-year period over a large region of Paris and its surrounding area.

The paper is structured as follows. In Sect. 2 the data used is described. The methods to derive LECDA and RECDA as well as the validation with in-situ Cloudnet data are explained in Sect. 3. Section 4 presents the results and discussion of the com-

parison as well as the validation of the two cloud masking approaches, and a meteorological case using RECDA. Section 5 contains the conclusions.

## 2   Data

### 2.1   SEVIRI satellite data

The main part of this study, the generation of cloud masks, is based on the analysis of SEVIRI HRV data that has a spatial

resolution of 1 km at nadir and covers a broadband range from 0.4 to 1.1 $\mu m$. Together with the remaining 3 solar and 8 thermal SEVIRI channels with a spatial sampling distance of 3 km at nadir, the Earth is observed in a repeat cycle of 15 minutes covering the full disc for the low-resolution and half of the full disc for the HRV channels (Schmetz et al. (2002), MSG Level 1.5 Image Data Format Description). For the detection and removal of snow and ice clouds channel reflectances of the visible (0.6 and 1.6 $\mu m$) and brightness temperatures of the infrared channels (8.7, 10.8 and 12.0 $\mu m$) are used, respectively

(see Sect. 3.1). In total the time period of the data set spans the month of November from 2004 to 2019 (08:00 - 16:00 UTC) resulting in $\sim$14400 time steps. The study region is the urban area of Paris and the adjacent rural areas extending from 48.0°N to 49.6°N and from 1.6°E to 3.0°E (Fig. 4b)) where urban cloud modifications have already been observed and considered





as relevant for the urban climate of Paris during summer (Theeuwes et al., 2019). The urban area of Paris serves as an ideal testbed as it is surrounded by relatively homogeneous and flat terrain with altitude ranges of ~200 metres (see Fig. 6b)). The

month of November was chosen to investigate a possible reduction of fog clouds over the urban area of Paris as found in other regions during winter (Gautam and Singh, 2018). Further, with the selection of one month, seasonal variations of land cover due to e.g. the vegetation period can be neglected and long-term variability of the surface reflectance can assumed to be small.

## 2.2   Cloudnet data

The cloud masks are validated against the calibrated Cloudnet classification product (CF-1.0, Level 2) for November (2015

to 2019) at Palaiseau (48.718°N – 2.202°E) located ~25km to the Southwest of Paris are used (Illingworth et al., 2007). The location of the Cloudnet station is shown in Fig. 2. Ground-based remote sensing instrumentation including a ceilometer, a doppler cloud radar and a microwave radiometer of the atmospheric observatory SIRTA (Site Instrumental de Recherche par Télédétection Atmosphérique, Haeffelin et al. (2005)) as well as model outputs of the ECMWF Integrated Forecast System (IFS) serve as input for the Cloudnet Target classification algorithms described in ACTRIS Deliverable WP5 / D5.5 (M24). As

a result a classification into 11 classes distinguishes ice and water clouds, precipitation, aerosols, insects, clear sky and combinations thereof (Appendix A: Table A1) is obtained in a 30s time interval and for 719 heights above sea level (168.5-18118.5m).

## 2.3   Reanalysis data

To highlight the applicability of the novel cloud mask data set generated by RECDA to study surface-driven modifications

of fog cloud cover, ERA5 data are used to filter for atmospheric conditions under which these clouds typically occur. ERA5 hourly data on single levels from 1979 to present of the European Centre for Medium-Range Weather Forecasts (ECMWF; (Hersbach et al., 2018) are used. The filters consider situations only if all of the following criteria are met: 10 m wind speed < 3 m s$^{-1}$ (U/V wind component), boundary layer height < 300 m, and mean sea level pressure > 1020 hPa. These are boundary layer conditions that characterize and facilitate the formation of fog and low stratus.

## 2.4   Corine land cover

For comparison of cloud fraction anomalies over different land cover types, Corine Land Cover (CLC) Level 3 data for 2012 (Version 2020_20u1) at 100 m spatial resolution provided by the Copernicus Land Monitoring Service is used (European Environment Agency, EEA (2021)). The land use in 2012 only shows minor differences compared to the available land cover classifications in 2006 and 2018 and is chosen here to represent the period 2004–2019. CLC 2012 data are resampled to the

HRV spatial resolution by using the most frequent land cover class within a HRV pixel (Fig. 4a)). Only relevant land cover classes including forests, continuous/discontinuous urban fabric, arable land and pastures are considered for a more detailed analysis.



## 2.5 Digital elevation model

The European Digital Elevation Model (EU-DEM; version 1.1) is used to compare changes in altitude to small-scale features
of cloud fraction anomalies (Fig. 6b)). The data set has a spatial resolution of 25 m and is provided by the Copernicus Land
Monitoring Service (EEA, 2017; Tøttrup, 2014).

## 3   Methods

### 3.1   Snow and Ice filter

The preprocessing of the HRV data includes the elimination of ice clouds and snow in order to pertain only liquid and mixed
clouds for the data analysis. This is done by using threshold tests on the SEVIRI channels at 0.6, 1.6, 8.7, 10.8 and 12 $\mu m$ and
collocating the flagged pixels with the associated HRV pixels. In this way, data that are not relevant for the analysis of urban
cloud modifications are excluded from the data set.

Snow is detected by calculating the Normalised Difference Snow Index (NDSI) based on the 0.6 and 1.6 $\mu m$ channels:

$$NDSI = \frac{r_{0.6} - r_{1.6}}{r_{0.6} + r_{1.6}} \tag{1}$$

where $r_\lambda$ is the reflectance at wavelength $\lambda[\mu m]$ (Dozier and Painter, 2004; Cermak, 2006). Tests done for specific scenes
revealed a threshold of 0.3 to be suitable to detect snow. Pixels that feature an NDSI above this value are excluded from the
data set. This approach has been widely and operationally applied to different satellite data including MODIS and Sentinel-2
satellite sensors (Hall et al., 2001; Richter et al., 2012).

A second filter is applied to exclude ice clouds from the data set with a brightness temperature below 263 K in the 10.8 $\mu m$
channel. To exclude potentially remaining ice clouds, the difference of the 8.7 and 12$\mu m$ channels is used in a phase test to
retain only liquid clouds. Ice absorbs much more strongly than liquid water between 10 and 13$\mu m$, while both show a similar
absorption pattern from 8 to 10$\mu m$ (Strabala et al., 1994; Cermak, 2006). The brightness difference should thus be smaller for
ice clouds and is implemented as in Westerhuis et al. (2020) with the fog and low stratus confidence level $CL_{FLS}$:

$$CL_{FLS} = \frac{T_{LC} - (T_{12\mu m} - T_{8.7\mu m}) - T_{CCR}}{-2xT_{CCR}} \tag{2}$$

where $T_{LC}$=1.8K is the threshold for liquid clouds and $T_{CCR}$=1K is the cloud confidence range. Details of this approach can
be found in Westerhuis et al. 2020. Pixel values with a $CL_{FLS}$ below 0 are removed from the dataset to retain only low-level
liquid clouds.

### 3.2   Two approaches to delineate clouds and clear sky

#### 3.2.1   Gaussian mixture model

Based on this preprocessed data set the Local Empirical Cloud Detection Approach and the Regional Empirical Cloud De-
tection Approach is proposed to delineate clouds from clear sky resulting in two different cloud masks (CMloc and CMreg).





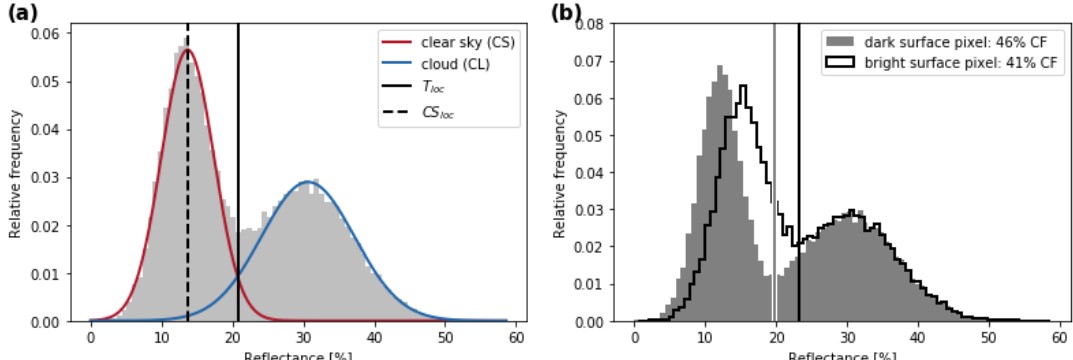

**Figure 1.** a) Conceptional distribution of a bimodal distribution of HRV pixel counts for one SZA bin and one pixel within the study region during November 2004–2019; 1st and 2nd component of the gaussian mixture model present clear (red) and cloudy conditions (blue) with the local clear sky reflectance $CS_{loc}$ (vertical black dashed line) and a potential clear-cloud threshold $T_{loc}$ (vertical black line). Clear sky reflectance is defined as the maximum count per histogram bin (maxbin) of the CS GMM component. The histogram minimum would be a potential threshold to separate the near-gaussian distributions of clear and cloudy pixels for this time and location. b) Same as (a) over a bright and a dark surface with the corresponding cloud fraction (CF). Both figures are created with 3 randomly generated gaussian distributions: a distribution for the cloudy conditions and two distributions for surfaces of different brightness (Sect. 3.2), showing the surface-brightness-driven shift in the threshold that leads to different estimates of cloud fraction in this artificial example.

The ultimate goal of RECDA is to derive spatially unbiased cloud fraction anomalies over different land cover types, while the goal of LECDA is a pixel-by-pixel cloud detection with minimum bias. Both approaches are relatively simple and nearly independent of other channel information. The thresholds are determined empirically, based on occurrence frequencies of HRV

counts in solar zenith angle (SZA) bins as shown conceptionally in Fig. 1a) for a single SZA bin and pixel. The first peak of the distribution presents clear sky conditions (CS) while the second refers to cloudy (CL) conditions. In both approaches the CS composite is obtained by applying a gaussian mixture model (GMM) to the HRV reflectance histograms of each SZA bin and each pixel within the study region. A GMM is a probabilistic model that assumes that the data are generated from a mixture of a finite number (here 2: CS and CL in 1a)) of Gaussian probability distributions with unknown parameters. This

method is categorized as an unsupervised learning algorithm that works in a similar way as the distance-based k-means cluster method. Instead of a distance-based model the algorithm finds the clusters based on Gaussian probability distributions. Based on the expectation-minimization algorithm, conditional expectations of the complete log-likelihood are calculated and estimates (mean and covariance) are updated iteratively until convergence to a local optimum is reached (Jain et al., 2000). While this method is limited to near-Gaussian distributions of a representative data set (excluding high SZA bins, where a bimodal

distribution of clear sky and cloud is not always existent) and a sufficient amount of data it has great potential beyond the simple usage of a binary cloud mask as it provides probabilities (Sci-kit learn implementation, Pedregosa et al. (2011)).
In this setup, especially the GMM characteristics of the CS component are exploited to dynamically (pixel-based per SZA bin)



derive thresholds. In the following, the computation of a local $T_{loc}$ (1) and a regional threshold $T_{reg}$ (2) is described based on the maximum bin count and standard deviation of the GMM CS component.

### 3.2.2 Local Empirical Cloud Detection Approach (LECDA)

In this approach a local HRV threshold $T_{loc}$ is obtained based on the CS probability distribution (1st GMM component) per SZA slot (5 SZA bins 67-77; >1000 data points) and per pixel as follows. First, the local clear sky reflectance $CS_{loc}$ (Fig. 1a)) is determined by the reflectance value with the maximum count per histogram bin (maxbin) with a binwidth of 0.5 within the $mean \pm 2\sigma$ range of the CS GMM component. Taking the maxbin approach is preferred over taking the mean of the 1st GMM component as it is representative for CS. Second, $T_{loc}$ is computed as the sum of $CS_{loc}$ (per pixel) and the regional median of all local $3\sigma$ of the CS GMM component. The regional median is chosen to increase the robustness of the derived $T_{loc}$ against a sub-optimal local fit of the GMMs due to atmospheric noise as e.g. aerosols. In general, this method is assumed to be less contaminated by the effect of cloud shadows, aerosols and thin cirrus clouds compared to similar clear-sky minimum approaches as it is based on the maximum bin count of the GMM clear-sky distribution (Bley and Deneke, 2013; EUMETSAT, 2019). However, the impact of these atmospheric features on the results cannot be entirely excluded and may distort the clear sky reflectance.

$$CS_{loc} = maxbin \in [mean + 2\sigma_{CSGMM}, mean - 2\sigma_{CSGMM}] \qquad (3)$$

$$T_{loc} = CS_{loc} + MED_{reg}(3\sigma_{CSGMM}) \qquad (4)$$

Thresholds obtained by LECDA as in Equ. 4 are used in e.g. Teuling et al. (2017) and present a straightforward method to approximate the "real" cloud fraction. In their study a similar local cloud mask approach is proposed where a constant threshold (ten counts) is added to an empirically determined clear sky value for a given pixel based on a cumulative distribution function of reflectivity measurements. However, such locally-optimized approaches including LECDA are dependent on variations in surface albedo due to different land cover types as a constant value is added to the clear sky reflectance shown in Fig. 2 and Equ. 4. A brighter clear-sky surface pixel within the urban area of Paris will shift the clear sky distribution to higher reflectance and thus clear-cloud thresholds assuming a constant cloud distribution as schematically illustrated using synthetic data in Fig. 1b). Due to the dynamic threshold method (pixel-based per SZA bin) that seeks to distinguish bright clear-sky surface pixels from clouds in a narrower bimodal distribution, a higher threshold is set in the case of brighter pixels compared to darker pixels. As a direct consequence, the application of LECDA for the retrieval of cloud fraction can potentially result in an underestimation of clouds over bright surfaces when compared to clouds over dark surfaces (example in legend of Fig. 1b)) given a constant cloud distribution over both surfaces.

Because the cloud detection results from LECDA may be subject to spatial biases related to spatial variations in surface albedo, the RECDA that uses a "static" clear-cloud threshold was developed (see Sect. 3.2.3).





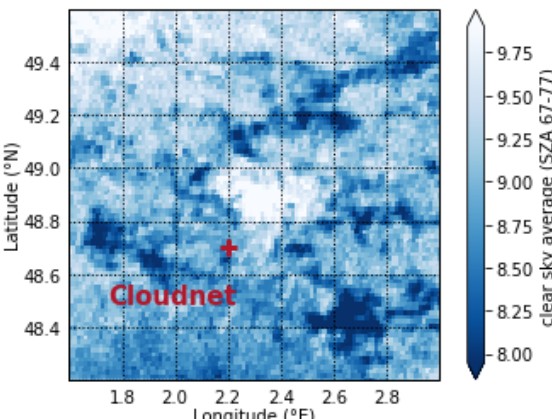

**Figure 2.** Average (SZA bins 67-77) of clear sky reflectance $CS_{loc}$ in [%] obtained by Equ. 3. The location of the Cloudnet station at Palaiseau used for validation is marked in red.

### 3.2.3 Regional Empirical Cloud Detection Approach (RECDA)

In this approach $T_{reg}$ is obtained by setting the maximum of all $T_{loc}$ as a "static" threshold that is applied to all pixels in the study area per SZA.

$$T_{reg} = max(T_{loc}) \tag{5}$$

RECDA is thought to result in a slight underestimation of clouds for all pixels as it sets the reflectance threshold to a higher reflectance value. The advantage compared to the LECDA is, however, that RECDA is independent of different surface albedo values (Fig. 2) and is therefore better suitable for the comparison of regional cloud fraction anomalies over different land surfaces.

### 3.3 Local evaluation using Cloudnet data

Cloudnet data are used to evaluate and compare the local performance of both approaches. In order to aggregate the Cloudnet Data to the temporal resolution of SEVIRI (15 minute interval), Cloudnet cloud fraction is calculated for each 15 minute time window (30 time steps). The time window of 15 minutes is approximated by the time a cloud needs to traverse the satellite reference pixel region of 3×3 pixels ($\sim$ 3×5 km). This is roughly estimated based on the length of the ground track ($\sim$ 4 km) times a length scale factor of 2 divided by the average wind speed of $\sim$ 10 m/s at cloud top height (Greuell and Roebeling, 2009).

The time window is centred around the corresponding SEVIRI time slot plus a time scan correction factor of 11 minutes. This correction factor was applied to match the SEVIRI nominal time with the actual scanning time over the validation site. Its calculation is based on the time SEVIRI needs per revolution (0.6 s) to scan 1 line (3 image lines at a time) from South to North accounting for the acquisition start at 81°S, the spreading distance of the ground resolution in S-N direction as well as





the number of scan lines (MSG Level 1.5 Image Data Format Description). The W-E scan time of 30 ms is neglected. Details of the computation for the low resolution channels can be found in (Kim et al., 2020) with the same nominal time as the HRV

channel.

All Cloudnet Target classifications per time window below an altitude of 3 km are considered as relevant for the validation of the cloud mask. To create a Cloudnet cloud fraction as ground truth, classes 1 to 7 are assigned as cloudy, while 0, and 8 to 10 as clear sky (cf. Tab. A1; Veefkind et al. (2016)). If the resulting Cloudnet cloud fraction of a time window is above 0.9, the flag of the corresponding time is assigned as cloud (1), otherwise clear sky (0). This threshold will ensure only persistent cloud

observations to be matched with the SEVIRI cloud mask.

The temporally aggregated Cloudnet cloud mask is compared to a spatial aggregate of 3×3 HRV cloud mask pixels centred around the cloudnet validation station. Due to the oblique viewing angle of SEVIRI a parallax correction is required resulting in a horizontally displacement of the pixel matrix surrounding the validation station by ∼2.6 km (2 pixel) to the South (Greuell and Roebeling, 2009). If more than one out of 9 matrix pixels are cloudy the matrix is aggregated to one value assigned as

cloud (1), otherwise clear sky (0). This matrix approach was chosen in order to fully capture clouds traversing the validation sight and to reduce the effect of cloud inhomogeneity and partial cloud cover (Deneke et al., 2005).

The performance of both approaches is measured by calculating the probability of detection (POD), false alarm rate (FAR), percent correct (PC), critical success index (CSI), bias score (BS), and the Heidke skill score (HSS) (equations in Appendix B).

## 230   4   Results and discussion

### 4.1   Comparison of the two cloud masking approaches

Fig. 3 shows cloud fractions and their regional anomalies from the Local and Regional Empirical Cloud Detection Approaches, CFloc and CFreg. Where CFloc and CFreg are defined as the number of cloudy pixels divided by the total number of pixels, and their regional anomaly ($CF_{loc}anomaly$, $CF_{reg}anomaly$) as the difference between the cloud fraction of the pixel and the

average cloud fraction of the region. The spatial patterns of $CF_{loc}$ and $CF_{reg}$ in Fig. 3a) and c) are similar and show a high cloud fraction and hence a positive anomaly in Fig. 3b) and d) in a triangular shape east and northeast of Paris where the river valleys of the Oise, Marne and Seine are located. While $CF_{loc}$ shows generally higher cloud fractions and a slightly larger anomaly range with more small-scale features, $CF_{reg}$ spatial patterns are less distinct and smoother. As expected due to the algorithm differences (see Sect. 3.2), $CF_{reg}$ is lower (on average by 7 %) than $CF_{loc}$.

Differences of the two cloud-masking approaches are shown in Fig. 4a) by mapping the difference of their respective cloud fraction anomaly patterns ($CF_{loc}anomaly$-$CF_{reg}anomaly$). The differences are most prominent over the urban region of Paris and over forests (cf. Fig. 4b) and clearly connected with the clear sky reflectance of these different land cover types (cf. Fig. 2). The comparison of the respective cloud fraction anomalies reveals a relative underestimation of $CF_{loc}$ by more than 4 % over the relatively bright urban area of Paris, while it leads to a relative overestimation of clouds over some of the

relatively dark forest areas. Grouping these cloud fraction anomaly differences by land cover types confirms these observations



Atmospheric


and shows that the relative biases over the continuous urban area of Paris exceeds the biases over the forest region (Fig. 5). The land cover type of pastures in the northwestern parts of the region also shows a relative underestimation of the local approach, however, local differences to the surrounding land cover types are not clearly visible.

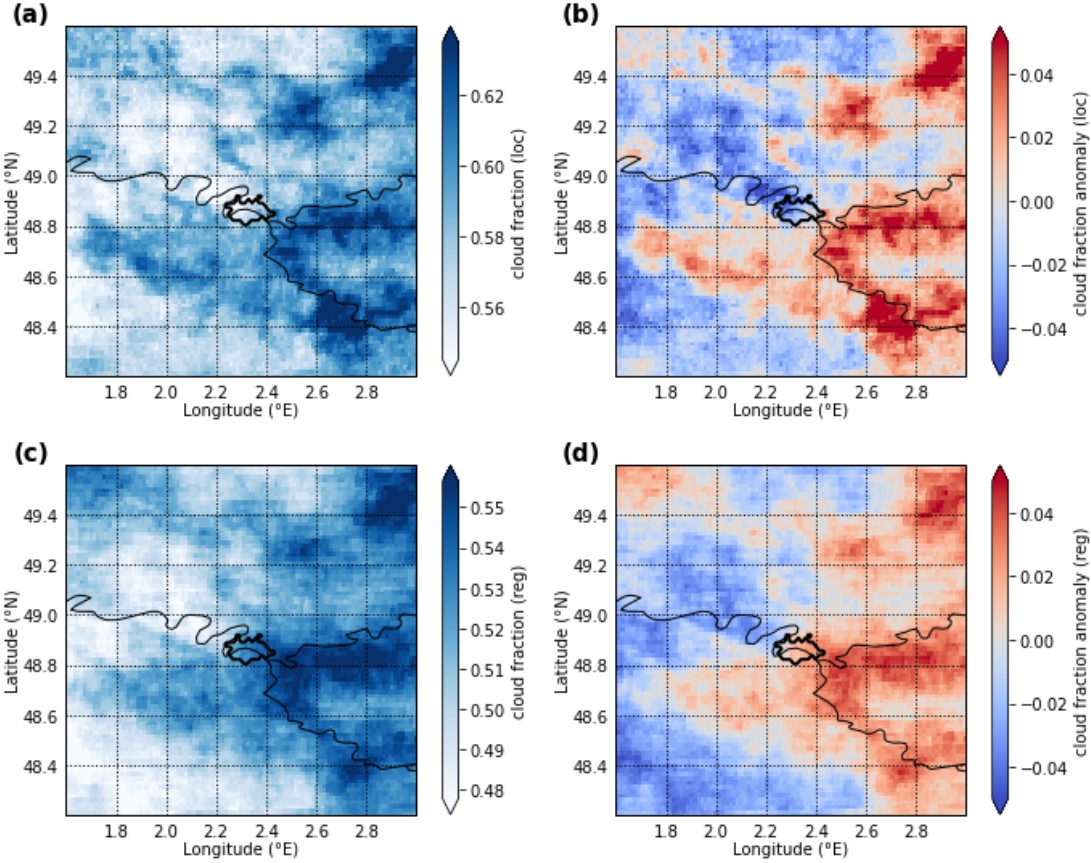

**Figure 3.** Cloud fraction $CF_{loc}$ and $CF_{reg}$ (a, c) as well as their spatial anomalies $CF_{loc}anomaly$ and $CF_{reg}anomaly$ (b, d) over the region of interest for the month of November during the years 2004-2019. Larger rivers are visualized as black lines, the continuous urban area of Paris as a black contour.

## 250    4.2    Analysis of cloud patterns in typical fog conditions

Patterns of cloud clearings over urban areas are mostly expected in conditions of fog or low stratus (Gautam and Singh, 2018). In order to test the application of $CM_{reg}$ to study spatial patterns of urban cloud modification, the derived cloud mask is filtered for specific boundary layer conditions using the ERA5 reanalysis data (cf. Sect. 2.3). Cloud fraction anomaly patterns constrained by these meteorological conditions show small-scale features that can be clearly associated with surface





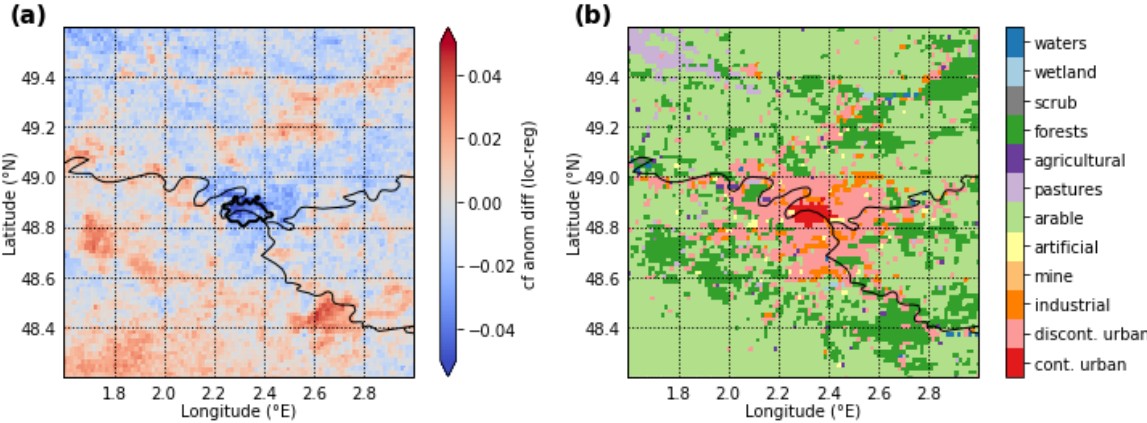

**Figure 4.** a) Difference of cloud fraction anomalies ($CF_{loc}anomaly$-$CF_{reg}anomaly$) from Local and Regional Empirical Cloud Detection Approaches. b) Main Corine Land Cover classes L2 and L3 for the year 2012. Larger rivers are visualized as black lines.

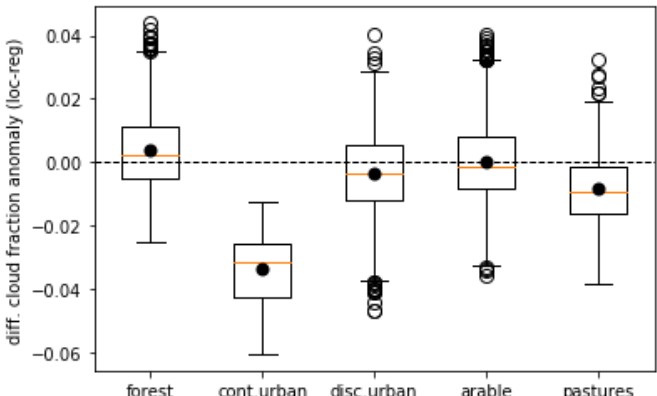

**Figure 5.** Difference of cloud fraction anomalies ($CF_{loc}anomaly$-$CF_{reg}anomaly$) from Local and Regional Empirical Cloud Detection Approaches grouped by 5 main Corine Land Cover types: forest, continuous urban fabric, discontinuous urban fabric, arable land and pastures. Median and mean of each class are presented as orange horizontal lines and black dots, respectively. Whiskers are equal to 1.5 times IQR beyond the first and third quartiles. Outliers are shown as black circles.

characteristics (Fig. 6). Notable is the distinct relative decrease of presumed fog and low stratus cloud fraction directly over the center of Paris and extending to its western edges in the order of ∼10 % relative to the regional average CF. This spatial pattern is usually associated with the dissipation of fog or low stratus by depletion of liquid water at the surface or the lifting of the cloud base (Wærsted et al., 2019; Williams et al., 2015; Underwood and Hansen, 2008; Haeffelin et al., 2005). This negative cloud fraction anomaly is likely a signal of the occurrences of fog holes (Gautam and Singh, 2018) over the urban center of Paris. It is interesting to note that the negative anomaly extends to the west of the city, suggesting that in the high pressure situations considered here, winds maybe predominantly from easterly directions.





Positive cloud fraction anomalies show clear links to topography especially over river valleys where fog is naturally favoured (Bendix, 2002). This is visible in the Northeast (Oise), the Northwest (Epte, Avelon, Thérain) and East and Southeast (Marne, Seine) of the study area.The high-resolution cloud fraction anomaly map thus highlights the ability of the regional approach to
detect small-scale cloud patterns that can be linked to land cover characteristics and the topography.

For a systematic assessment of the land-surface influences on spatial cloud fraction anomalies in the study region, cloud fraction anomalies are grouped by land cover types in Fig. 7. The land cover class "continuous urban" shows an average negative cloud fraction anomaly of 6 % that can be interpreted as a lower bound of the magnitude of the city's effect on fog occurrence, as Paris is located in a river valley and should thus have favourable conditions for fog formation. Over pastures an increase by
4 % is notable, however, this is likely linked to the location of the pastures in the Avelon river valley in the NW of the study area. Smaller anomalies of the remaining land cover classes are partly explained by the combined influence of land cover type, terrain height and the dynamic of the fog hole evolution itself. The way this cloud pattern is linked to the urban heat island of Paris, land surface characteristics and further meteorological variables will require further analysis beyond the scope of this paper.

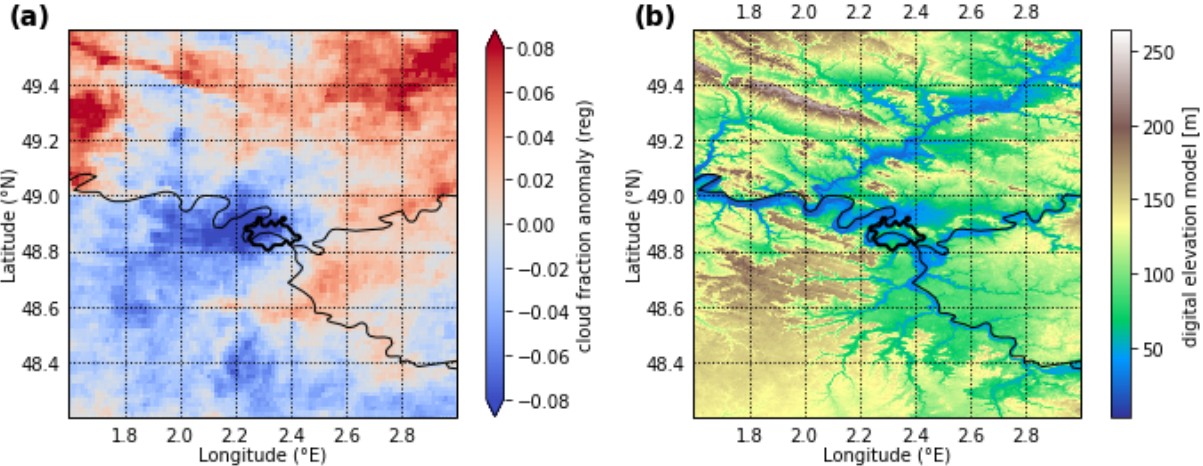

**Figure 6.** a) Cloud fraction $CF_{reg}$ anomaly from the Regional Empirical Cloud Detection Approach constrained by the following meteorological conditions: low wind speed (<3m/s), low blh (<300m), high msl (>1020hPa). b) European Digital Elevation Model with a spatial resolution of 25m. The Seine river is visualized as black line.

### 4.3 Validation with ground-based cloud fraction

The validation of the two cloud masks $CM_{loc}$ and $CM_{reg}$ in Tab. 1 shows that both approaches perform similar in terms of overall quality (cf. HSS: 0.69 and 0.71; CSI: 0.72), but achieve this with very different performance characteristics. In general, $CM_{loc}$ outperforms $CM_{reg}$ with a probability of detection (POD) of 0.95 compared to 0.86. However, the false alarm ratio (FAR) of 0.18 for $CM_{reg}$ falls below that for $CM_{loc}$ of 0.25. These results are expected as $T_{loc}$ is generally lower than $T_{reg}$



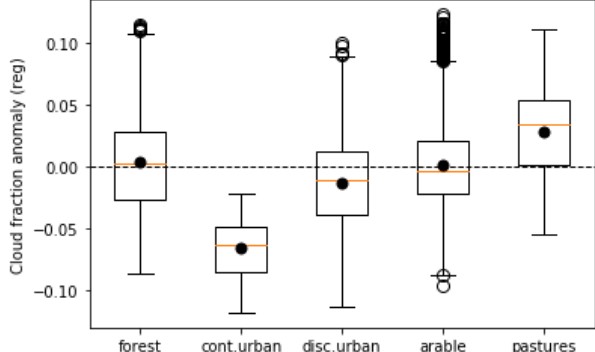

**Figure 7.** Cloud fraction $CF_{reg}$ anomaly from the Regional Empirical Cloud Detection Approach constrained by meteorological conditions (Fig. 6) and grouped by 5 main Corine Land Cover types: forest, continuous urban fabric, discontinuous urban fabric, arable land and pastures. Details as in Fig. 5.

(except where $T_{loc} = \max(T_{loc})$), leading to a general underestimation of clouds in $CM_{reg}$ with the benefit of reducing false alarms.

The Cloudnet data validate both approaches over a location with a surface reflectance of 8.65 % (see Fig. 2) that is close to the scene average of 8.96 %. It is expected that the validation of the local and regional approach will give the same or nearly identical results over the bright surfaces of the city, while over darker forest regions, the local approach is likely to have a much
higher POD, with little influence on the FAR.

For assessing the performance of both approaches (LECDA and RECDA) in the context of various thresholds, POD and FAR are computed along the range of HRV values and exemplary for different SZA (Fig. 8). The resulting pseudo-ROC (receiver operating characteristic) curves can be used to determine the ideal threshold for cloud detection, which typically can be found in the upper left-hand corner with a minimal FAR and a maximal POD. The main outcome is that $CM_{reg}$ for the SZA bins
67(–69) and 69(–71) is close to being an optimal tradeoff, achieving a low FAR while loosing only little detection skill (POD) when compared to $CM_{loc}$ and alternative HRV thresholds. The proposed regional approach, however, shows a lower POD compared to the local approach that can be attributed to the general cloud underestimation that is true for all the different surface reflectances equally.

The ROC curves further show a degradation of performance in SZA bins > 71 in both approaches. This SZA dependence of
surface albedo and thus clear sky reflection is expected due to the gradual approximation of cloud and clear sky reflectance in the morning and evening twilight and known by other cloud mask algorithms as well, e.g. EUMETSAT (2019). Due to the similar signature of clouds and clear skies the number of false alarms is increased while the POD decreases. The degradation of performance towards lower SZA (not shown) is explained by the limited amount of data that is available for the lowest SZA bins. In general, the month of November, though relevant for this study, has only a limited number of daytime hours and thus
available data per SZA slot.

The ground-based validation of the satellite-retrieved cloud masks is difficult as differences can be due to the specific measure-





ment methods, including the different scales and the way cloud net data is aggregated to match the satellite data. Finding the optimum scales of cloudnet averaging to match satellite observations is complex (cf. Greuell and Roebeling (2009)).

**Table 1.** Cloudnet validation results for the cloud masks using the local ($CM_{loc}$) and the regional ($CM_{reg}$) threshold. Validation measures: probability of detection (POD), false alarm ratio (FAR), percentage correct (PC), critical success index (CSI), bias score (BS) and Heidke skill score (HSS).

| Class | POD | FAR | PC | CSI | BS | HSS |
|---|---|---|---|---|---|---|
| $CM_{loc}$ | 0.95 | 0.25 | 0.84 | 0.72 | 1.27 | 0.69 |
| $CM_{reg}$ | 0.86 | 0.18 | 0.86 | 0.72 | 1.05 | 0.71 |

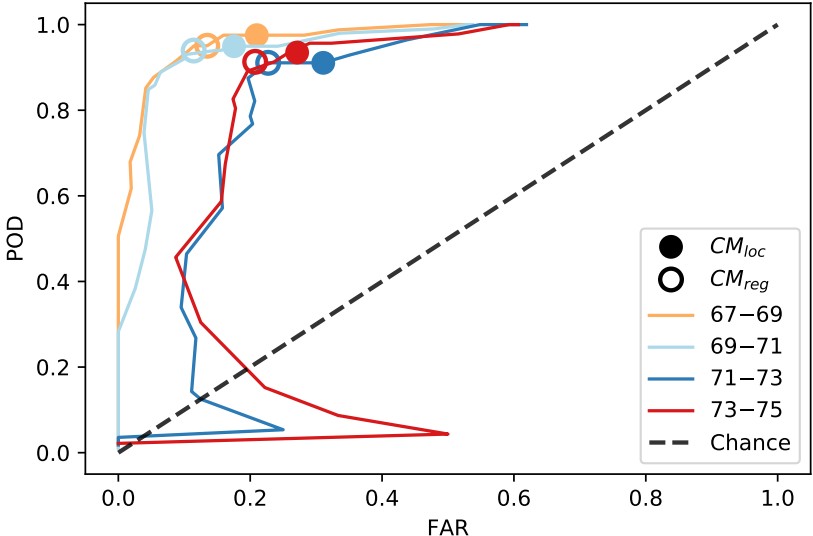

**Figure 8.** Relationship of POD and FAR for local and regional cloud masks ($CM_{loc}$ and $CM_{reg}$; filled and transparent circles) and for a range of different HRV threshold values and four Solar Zenith Angle (SZA) bins (colored lines). HRV threshold values increase from top to bottom of the plot, from HRV minimum to HRV maximum per SZA. The pseudo-ROC curve shows the best performance of the HRV threshold values used for cloud masking in the upper left corner, while accuracy decreases towards the dashed diagonal.



## 5 Conclusions

This study presents and compares the applicability of two empirical cloud masking approaches based on the single HRV channel of MSG SEVIRI for the high-resolution analysis of land surface effects on boundary layer clouds. The performance of the cloud mask based on the regional approach RECDA is compared to the local approach LECDA with respect to the influence of the underlying surface. Both cloud mask approaches and obtained cloud fractions are analysed over the ∼150×150 km region centered on Paris, where holes in fog and low stratus are expected.

It is shown that cloud masks obtained from LECDA result in a relative underestimation of cloud occurrence over the bright urban surface of Paris (up to 5 %), while cloud occurrence is overestimated in relative terms over dark surfaces as e.g. forests (up to 5 %) when compared to RECDA. This leads to the conclusion that studies using such locally-optimized approaches have to be attentive when associating different land surface classes to cloud occurrence.

The application of the regional approach RECDA to compute cloud masks in contrast to a pixel-based surface-dependent threshold is shown to be advantageous as local biases due to differences of surface reflectance can be prevented. As suggested in the study's context the surface-independent cloud mask obtained from empirically based regional thresholds is robust towards variability of the surface reflectance and show a reduced FAR. Based on this approach a reduction of cloud cover filtered for typical fog/low stratus conditions up to 10 % could have been observed for the month of November over and to the west of the urban area of Paris. This spatial cloud pattern is associated with the urban surface that probably affects the boundary layer through gradual heat release by the urban surface. In addition to this prominent decrease small-scale cloud enhancements attributed to river valleys, topography and forests regions prove the plausibility of the cloud mask. It should be noted that the cloud masks obtained with RECDA are likely underestimating cloud occurrence so that their application is limited to the quantification and analysis of regional cloud cover anomalies.

This study shows the great potential of a HRV-derived robust, land-surface independent cloud mask for the satellite-based quantification of small-scale interactions of the urban surface and boundary layer clouds in the range of ∼1-2 km. The relatively simple and independent implementation and application of this approach for regional and urban analyses is expected to have a diverse potential on an Europe-wide scale.



# Appendix A: Definition of Cloudnet Target classes

**Table A1.** Cloudnet target classification

| Class | Definition |
|---|---|
| 0 | Clear sky |
| 1 | Cloud liquid droplets only |
| 2 | Drizzle or rain |
| 3 | Drizzle or rain coexisting with cloud liquid droplets |
| 4 | Ice particles |
| 5 | Ice coexisting with supercooled liquid droplets |
| 6 | Melting ice particles |
| 7 | Melting ice particles coexisting with cloud liquid droplets |
| 8 | Aerosol particles, no cloud or precipitation |
| 9 | Insects, no cloud or precipitation |
| 10 | Aerosol coexisting with insects, no cloud or precipitation |



## Appendix B: Equations of statistical validation measures

$$POD = \frac{a}{a+c} \tag{B1}$$

$$FAR = \frac{b}{a+b} \tag{B2}$$

$$PC = \frac{a+d}{a+b+c+d} \tag{B3}$$

$$CSI = \frac{a}{a+b+c} \tag{B4}$$

$$BS = \frac{a+b}{a+c} \tag{B5}$$

$$HSS = \frac{2(ad-bc)}{(a+c)(c+d)+(a+b)(b+d)} \tag{B6}$$

with a = number of hits, b = number of false alarms, c = number of misses and d = number of correct negatives.



*Author contributions.*  JF fully developed the concept and methodology and wrote the software, obtained and analyzed the data sets, conducted the original research and wrote the manuscript. JF, HA and JC discussed and improved the algorithm. EP preprocessed the CLC data.
RR contributed to the validation of the algorithm. HA contributed to the interpretation of the results. JF, HA, JC, EP and RR reviewed and edited the manuscript.

*Competing interests.*  The authors declare that they have no conflict of interest.

*Acknowledgements.*  We acknowledge the use of CloudNet data which are part of the European Aerosol, Clouds and Trace Gases Research Infrastructure (ACTRIS) project. The research leading to these results has received funding from the European Union's Horizon 2020 research
and innovation programme under grant agreement No 654109 and the Cloudnet project (European Union contract EVK2-2000-00611) for providing the Cloudnet classification product, which was produced by the Department of Meteorology, University of Reading using measurements from the atmospheric observatory SIRTA at Palaiseau. ERA5 data (Hersbach et al., 2018) were obtained from the Copernicus Climate Change Service (C3S) Climate Data Store. The European Environment Agency (EEA) is acknowledged for the CORINE land cover (https://land.copernicus.eu/pan-european/corine-land-cover) and the EU-DEM (https://land.copernicus.eu/imagery-in-situ/eu-dem).
The results contain modified Copernicus Climate Change Service information 2020. Neither the European Commission nor ECMWF is responsible for any use that may be made of the Copernicus information or data it contains. Eva Pauli has been financially supported by the Graduate Funding from the German States. We acknowledge support by the KIT-Publication Fund of the Karlsruhe Institute of Technology.



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
