# Peer review of "High-resolution satellite-based cloud detection for the analysis of land surface effects on boundary layer clouds"

_Atmospheric Measurement Techniques, 2022_

## Referee Comment (RC1)

**Review AMT Manuscript amt-2022-36 entitled High-resolution satellite-based cloud detection for the analysis of land surface effects on boundary layer clouds**

**General comments**

This article describes an approach for using high resolution satellite data to investigate land surface driven modifications of cloud occurrence. Two cloud masking algorithms are introduced, one local using pixel-based thresholds with a very high cloud detection skill and another one with a slightly smaller cloud detection skill but with the feature of being relatively independent of variations in the surface reflection. Both are based on the high-resolution visible channel of MSG SEVIRI which allows to study very small-scale cloud responses/adjustments to land surface changes. In my opinion, this topic is highly relevant to a wider community, as the interaction between biosphere and atmosphere comes more and more into focus. The presented approach in this study provides a tool for e.g., studying the effect of deforestation/forest decline, drought or urban expansion on cloud clover and thus the radiative balance. The article is generally very well written. However, some aspects could be explained a bit more detailed, especially the conclusions of the results and what the study means for future research, which are listed below. All of my comments are more or less text edits and therefore, no major changes are required in my opinion.

**Abstract:**

The first 8 lines are more like an introduction and motivation. The sentence line 9-10 is good and should come earlier in the abstract. Afterwards you go into too much detail in my opinion. I would not mention the Heidke Skill Score here (too much technical details) but focus more on the added value of the regional cloud mask algorithm in order to study how cloudiness is influenced by land surface type and albedo. I think that if the abstract includes some more results or conclusions (e.g. that with the regional cloud mask you can confirm the city's effect on fog dissipation).

**Conclusions:**

Some ideas which could be addressed in future:

What could be expected if the tool is applied to other regions of the world (raising cities, forest stress…)?

How could this method be applied in synergy with future satellite missions (e.g. FLEX and Biomass)? When we have knowledge on the state of the surface (forest health etc.), we want to know how this effect the occurrence of clouds. This could be a point to mention for future research on your topic.

**Detailed comments:**

*Line 41: small scale ~3 km features…*

> The effective resolution of the pixels is smaller due to two reasons. First, due to the viewing angle, the pixel area is more something like (3.1x6.1 km2), which is addressed at a later stage of the paper, when the parallax correction is described. The second effect is that the real optical resolution of MSG is lower as characterized by the modulation transfer function (MTF) and the pixels are oversampled in the image rectification process by a factor of about 1.6 (Deneke and Roebeling, 2010).

Thus the effective area of a pixel is slightly larger than the actual sampling resolution and this makes the features bigger as well.

*Line 67: …unbiased with respect to surface properties.*
maybe it becomes clearer when: is independent of variations in the surface properties, like spectral albedo. In one sentence, it could be added what is the gain of the development of another cloud detection algorithm, after all the different approaches are well described in the introduction.

*Line 68-72:*
I would mention here in one sentence the reason for choosing the month November and that the results are validated against a Cloudnet station. It comes in the next paragraph, but this is more an overview of the structure of the paper.

*Line 95 and other places:*
25km → The unit is often directly behind the value. I think that there should be half a space in between.

*Line 101: 719 heights*
What is the vertical resolution of the Cloudnet data, how important is the vertical resolution for your study? Or is it just mentioned to introduce Cloudnet in general?

Line 104 ff.
Could you provide some more details how you apply ERA5 data on Meteosat scenes? Do you reproject ERA5 data onto MSG or do you rather use the four criteria to be valid in your whole box around Paris?

*Line 115: … most frequent land cover class…*
Do you exclude cases, if the land cover class variability is too high? For your 2d maps, taking the most frequent one might be fine, but for your cloud fraction statistics, maybe exclude cases if "most frequent" is less than 50%? But maybe this is not really relevant and just a detail which does not need further attention. (Just thinking on sub grid scale variability like we have when comparing MSGs low resolution channels to 3x3 HRV pixels…)

*Line 186:*
Fig. 1b) → the bracket can be removed

*Line 195:*
How would you deal with RECDA, if T_loc max is a very bright artefact? Maybe a snow surface which is not detected by the snow flagging? Would it be maybe better

to have a compromise in between that would be maybe the 95$^{th}$ percentile of T_loc, but still being independent of land surface anomalies?

Line 199:

*Line 206:*
As mentioned in line 41, maybe take into account the pixel oversampling factor of 1.6?

*Line 216: below an altitude of 3 km*
What is the reason that you use the target classification only below 3 km?

*Line 218: Cloudnet cloud fraction is above 0.9*
How is this in relation to line 224 "If more than 1 out of 9 are cloudy the matrix…"? You justify that high threshold for Cloudnet to ensure only cloud persistent cases, but if a 3x3 HRV pixel is cloud contaminated, the low-resolution cloud mask will likely give "cloudy" as well. I am not familiar enough with cloud mask validation against Cloudnet, but intuitively I would consider cloud fractions also above 0.5 as cloud for the scene.

Line 261: from easterly direction
In your ERA5 criteria list, you don't have the wind direction? But I would assume that especially because of the location of Paris and the rivers around, the wind direction plays a significant role as well. Or can this be neglected when using winds below 10 m/s only?

*Line 266-274:*
This is an interesting analysis of the different cloud fraction anomalies, although not every result is very significant (CF increase over pastures). It would be very interesting to investigate the CF anomalies for land surface changes over years. I made some examples before (deforestation, city expansion, drought). If I get it right, you limit the analysis on the spatial variability, not the temporal. You state that the multi-year period makes the results robust under the assumption that the long-term surface variability is small. I find it very interesting to understand how human made surface modifications impact the cloud occurrence.

*Fig. 6: blh and msl*
Boundary layer height and mean sea level are mentioned but the acronym is not introduced before

*Line 289: The main outcome is that…*
This could be something mentioned in the abstract in other words.

*Line 319: over and to the west*
Better: over the urban area of Paris and west of it.

---

## Referee Comment (RC2)

AMThttps://doi.org/10.5194/amt-2022-36
High-resolution satellite-based cloud detection for the analysis of
land surface effects on boundary layer clouds by Julia Fuchs, Hendrik Andersen, Jan Cermak,
Eva Pauli, and Rob Roebeling

This paper examines two-slightly different regional cloud mask algorithms using the high-resolution broadband visible channel from MSG SEVIRI instrument for a region in Paris and its vicinity. Both algorithms started with pixel-level and solar zenith angle binned histogram of reflectance. The localized algorithm (LECDA) uses localized (down to pixel-level) thresholds based on gaussian fitted reflectance histogram, while the regional algorithm (RECDA) uses the maximum of LECDA thresholds for the entire domain. The RECDA algorithm is considered as independent of surface albedo.

The paper claims slightly better performance of RECDA method compared with measurements of a ground Cloudnet station. In addition, the cloud fraction from RECDA algorithm is shown to be able to demonstrate the impact of urban heat effect on fog formation from the city of Paris. The study is interesting as high-resolution cloud detection algorithm will enable the study of impact on cloud formation due to various natural and anthropogenic factors in very small scales. However, I am not totally convinced that RECDA is a better algorithm than the LECDA algorithm due to the following reasons:

1) Both algorithms have pros and cons as demonstrated by better POD, larger FAR in LECDA and poor POD, lower FAR in RECDA since the threshold in RECDA is higher than most used in LECDA. The large contrast in POD and FAR of the two algorithms and relatively insignificant difference in overall scores (PC, CSI, HSS) indicate that more fundamental difference of the two algorithm lies in the choice of a more clear-conservative or cloud-conservative approach rather than whether fine tuning of local threshold is better or worse. Therefore, what is more important in this case will depend on the application. Does the application require to have high POD or low FAR or an overall better score?

2) The LECDA aims to follow the reflectance distribution of clear sky pixels, as the algorithm is derived from the clear sky portion of the Gaussian distribution, while RECDA aims to preserve the cloud distribution as it assumes the cloudy portion of the Gaussian distribution does not change with surface albedo. This assumption will be more appropriate for thick clouds but not thin clouds as surface reflectance could also impact cloudy sky reflectance in the later conditions.

3) The evaluation is only conducted over one location even though the reflectance of the selected location is close to the domain mean. As mentioned by the author, the comparison over bright surface would be similar but over dark surface, LECDA is expected to have higher POD and little change in FAR. Therefore, over the entire

domain, it is yet to be seen which algorithm performs better. It may help to compare the cloud mask with other multi-channel satellite cloud mask products such as those from MODIS with 1km resolution and full spatial coverage.

4) The Cloudnet cloudy sample requires 90% of cloud fraction which is very cloud conservative while SEVIRI cloud masks only require 1/9 fraction to be cloud. This mismatch in spatial/temporal cloud fraction could contribute to slightly better performance of relatively underestimating RECDA algorithm as cloud detection rate could be even lower if more partial Cloudnet pixels are selected as cloudy.

5) The relative performance of RECDA and LECDA might change with the domain size and dominant cloud types in the region. It is well known that a globally fixed threshold does not work well.  How does the domain size and surface homogeneity (range of albedo) affect the performance of the two algorithms, especially RECDA?

Minor comments:
What is the bandwidth of the HRV of SERVIRI? Some website mentions 0.4-0.9 µinstead of 0.4-1.1 µm.

What is x in Equ. 2?

P7L185-190. I don't see how RECDA would not create surface albedo dependent bias unless the algorithm only focused on thick clouds (in that case surface albedo doesn't matter). It seems to be a cloud-conservative approach and assumes that reflectance distribution of cloudy pixels does not change. However, due to the overlap of clear sky and cloud sky histogram, fixing threshold for cloudy pixel distribution inevitably affects cutoff of clear sky distribution.

Figure 6. How is the anomaly computed? Are the anomalies computed with respect to domain averaged mean? Could you plot the same figure (Fig.6a) from the LECDA method?

---

## Author Comment (AC1)

**Response to interactive comment of referee 1 — High-resolution satellite-based cloud detection for the analysis of land surface effects on boundary layer clouds**

Julia Fuchs[1,2], Hendrik Andersen[1,2], Jan Cermak[1,2], Eva Pauli[1,2], and Rob Roebeling[3]

[1]Karlsruhe Institute of Technology (KIT), Institute of Meteorology and Climate Research, Karlsruhe, Germany
[2]Karlsruhe Institute of Technology (KIT), Institute of Photogrammetry and Remote Sensing, Karlsruhe, Germany
[3]European Organisation for the Exploitation of Meteorological Satellites (EUMETSAT), Darmstadt, Germany

We would like to thank the reviewer for the valuable and very constructive comments and the effort that was put into reviewing the manuscript. We think the scientific discussion helped improving the manuscript, substantially. Please find the point-to-point reply below.

**1 General comments**

5    This article describes an approach for using high resolution satellite data to investigate land surface driven modifications of cloud occurrence. Two cloud masking algorithms are introduced, one local using pixel-based thresholds with a very high cloud detection skill and another one with a slightly smaller cloud detection skill but with the feature of being relatively independent of variations in the surface reflection. Both are based on the high resolution visible channel of MSG SEVIRI which allows to study very small-scale cloud responses/adjustments to land surface changes. In my opinion, this topic is highly relevant to

10    a wider community, as the interaction between biosphere and atmosphere comes more and more into focus. The presented approach in this study provides a tool for e.g., studying the effect of deforestation/forest decline, drought or urban expansion on cloud clover and thus the radiative balance. The article is generally very well written. However, some aspects could be explained a bit more detailed, especially the conclusions of the results and what the study means for future research, which are listed below. All of my comments are more or less text edits and therefore, no major changes are required in my opinion.

15 **Abstract:**

The first 8 lines are more like an introduction and motivation. The sentence line 9-10 is good and should come earlier in the abstract. Afterwards you go into too much detail in my opinion. I would not mention the Heidke Skill Score here (too much technical details) but focus more on the added value of the regional cloud mask algorithm in order to study how cloudiness is influenced by land surface type and albedo. I think that if the abstract includes some more results or conclusions (e.g. that with

20    the regional cloud mask you can confirm the city's effect on fog dissipation).

Thanks for the recommendations. We modified the abstract according to your suggestions and added more conclusions. We think it is more concise now.

**Conclusions:**

Some ideas which could be addressed in future: What could be expected if the tool is applied to other regions of the world (raising cities, forest stress...)? How could this method be applied in synergy with future satellite missions (e.g. FLEX and Biomass)? When we have knowledge on the state of the surface (forest health etc.), we want to know how this effect the occurrence of clouds. This could be a point to mention for future research on your topic.

5    We have added these discussion points and concluding remarks in the conclusion and discussion part of the manuscript.

**2   Detailed comments:**

Line 41: small scale 3 km features...

The effective resolution of the pixels is smaller due to two reasons. First, due to the viewing angle, the pixel area is more

10   something like (3.1x6.1 km2), which is addressed at a later stage of the paper, when the parallax correction is described. The second effect is that the real optical resolution of MSG is lower as characterized by the modulation transfer function (MTF) and the pixels are oversampled in the image rectification process by a factor of about 1.6 (Deneke2010a). Thus the effective area of a pixel is slightly larger than the actual sampling resolution and this makes the features bigger as well.

We have added these relevant details to the Data section. However, "small scale 3 km features" refers to the HRV channel

15   resolution and a HRV pixel would cover a pixel area of 1x1.9km2 in our study region.

Line 67: ...unbiased with respect to surface properties. maybe it becomes clearer when: is independent of variations in the surface properties, like spectral albedo. In one sentence, it could be added what is the gain of the development of another cloud detection algorithm, after all the different approaches are well described in the introduction.

We modified the sentence accordingly and added another to highlight the gain due to the proposed algorithm.

20   Line 68-72:

I would mention here in one sentence the reason for choosing the month November and that the results are validated against a Cloudnet station. It comes in the next paragraph, but this is more an overview of the structure of the paper.

We have added the information here and think that this will increase readability.

Line 95 and other places:

25   25km → The unit is often directly behind the value. I think that there should be half a space in between.

Yes, thanks. Done.

Line 101: 719 heights

What is the vertical resolution of the Cloudnet data, how important is the vertical resolution for your study? Or is it just mentioned to introduce Cloudnet in general?

30   The vertical resolution is 25 m. If one of the vertical layers is classified as cloud, the time step is flagged as cloud. As a consequence, we think that the vertical resolution is sufficient to not miss out a cloud layer. The method section was modified for clarity.

Line 104 ff.

Could you provide some more details how you apply ERA5 data on Meteosat scenes? Do you reproject ERA5 data onto MSG or do you rather use the four criteria to be valid in your whole box around Paris?

Thanks for this remark. The ERA5 data was averaged over the study region and the SEVIRI data was filtered according to the ERA5 mean values meeting the defined criteria. We have added this information to the manuscript (see subsection "Reanalysis data").

Line 115: ... most frequent land cover class...

Do you exclude cases, if the land cover class variability is too high? For your 2d maps, taking the most frequent one might be fine, but for your cloud fraction statistics, maybe exclude cases if "most frequent" is less than 50 %? But maybe this is not really relevant and just a detail which does not need further attention. (Just thinking on sub grid scale variability like we have when comparing MSGs low resolution channels to 3x3 HRV pixels...)

Thanks for this suggestion. We are not considering the land cover class variability as we are expecting only a marginal effect on the cloud statistics. We have added this potential influence of this sub grid scale variability of the land cover classes within a HRV pixel as discussion point to the result section.

Line 186:

Fig. 1b) → the bracket can be removed

Done.

Line 195:

How would you deal with RECDA, if $T_{loc}$ max is a very bright artefact? Maybe a snow surface which is not detected by the snow flagging? Would it be maybe better to have a compromise in between that would be maybe the 95th percentile of $T_{loc}$, but still being independent of land surface anomalies?

Thanks for pointing this out. This is something we discussed internally before. It is true that a bright artefact can decrease the performance of the cloud mask (true clear sky classified as cloud) if it is occurring frequently and we have added the limitation to the manuscript. For a less frequent feature as e.g. not detected snow we would expect only a minor influence on the PDFs. Using a 95th percentile of $T_{loc}$ compared to the maximum value as a threshold in RECDA may lead contrary to LECDA to an artificial overestimation of clouds over the urban area. To remove such a surface-reflectance based bias as far as possible max($T_{loc}$) was therefore selected. Generally, the plausibility of the clear sky map is and should be tested for other study regions as well. If artefacts are obvious, the pixel should be neglected.

Line 199: Delete "however"

Done.

Line 206:

As mentioned in line 41, maybe take into account the pixel oversampling factor of 1.6?

We have added these relevant details to the Data section.

Line 216: below an altitude of 3 km

What is the reason that you use the target classification only below 3 km?

Initially we selected data below an altitude of 3 km to filter out high ice clouds. Now we think it is more consistent if we use

all altitudes for validation as ice clouds have been removed from SEVIRI as well. The validation was recalculated accordingly.

Line 218: Cloudnet cloud fraction is above 0.9

How is this in relation to line 224 "If more than 1 out of 9 are cloudy the matrix..."? You justify that high threshold for Cloudnet to ensure only cloud persistent cases, but if a 3x3 HRV pixel is cloud contaminated, the low-resolution cloud mask will likely give "cloudy" as well. I am not familiar enough with cloud mask validation against Cloudnet, but intuitively I would consider cloud fractions also above 0.5 as cloud for the scene.

Decreasing the cloud net cloud fraction or increasing the SEVIRI cloud fraction required to be classified as cloud will decrease POD together with FAR for both algorithms, RECDA and LECDA. Less clouds will be detected in the cloud mask and less false alarms will occur. Finding the optimum aggregation scale is complicated as we are comparing a vertically and temporally highly resolved point measurements of Cloudnet with a 15 minute SEVIRI snapshot of the cloud top over a larger spatial area. Following the recommendations of reviewer 2 and 1 we now decided using more moderate thresholds. We recalculated the validation measures for both algorithms using a Cloudnet cloud fraction of 50% over a time window of 1 hour (Deneke et al. 2009) and a SEVIRI cloud fraction of 4/9. The results show a degradation of the performances of both RECDA and LECDA with respect to POD, while FAR is reduced. We have modified the manuscript accordingly.

Line 261: from easterly direction

In your ERA5 criteria list, you don't have the wind direction? But I would assume that especially because of the location of Paris and the rivers around, the wind direction plays a significant role as well. Or can this be neglected when using winds below 10 m/s only?

The purpose of the selection was mainly aiming at typical fog conditions to test the application of the cloud mask. This could be achieved with the listed criteria. We assume that wind direction plays a minor role for the definition of typical fog conditions as we are filtering for very calm conditions (<3 m/s). We have added this explanation to the manuscript. We think that a stratification according to the wind direction may presumably slightly shift the area of the cloud decrease that is related to Paris to a specific direction. This is something we would like to investigate further but would require a more profound analysis outside the scope of the study.

Line 266-274:

This is an interesting analysis of the different cloud fraction anomalies, although not every result is very significant (CF increase over pastures). It would be very interesting to investigate the CF anomalies for land surface changes over years. I made some examples before (deforestation, city expansion, drought). If I get it right, you limit the analysis on the spatial variability, not the temporal. You state that the multi-year period makes the results robust under the assumption that the long-term surface variability is small. I find it very interesting to understand how human made surface modifications impact the cloud occurrence.

Yes we agree this would be a very interesting direction for further research. We would have to test the performance of the cloud mask with respect to long-term surface changes. The thresholds should be independent of temporal surface signal variations as well. We have added this potential application to the conclusions.

Fig. 6: blh and msl

Boundary layer height and mean sea level are mentioned but the acronym is not introduced before

The acronyms are now introduced.

Line 289: The main outcome is that. . .

This could be something mentioned in the abstract in other words.

5   We modified the abstract.

Line 319: over and to the west

Better: over the urban area of Paris and west of it.

Done.

**References**

Deneke, H. M., Knap, W. H., and Simmer, C.: Multiresolution analysis of the temporal variance and correlation of transmittance and reflectance of an atmospheric column, Journal of Geophysical Research Atmospheres, 114, https://doi.org/10.1029/2008JD011680, 2009

---

## Author Comment (AC2)

**Response to interactive comment of anonymous referee 2 — High-resolution satellite-based cloud detection for the analysis of land surface effects on boundary layer clouds**

Julia Fuchs[1,2], Hendrik Andersen[1,2], Jan Cermak[1,2], Eva Pauli[1,2], and Rob Roebeling[3]

[1]Karlsruhe Institute of Technology (KIT), Institute of Meteorology and Climate Research, Karlsruhe, Germany
[2]Karlsruhe Institute of Technology (KIT), Institute of Photogrammetry and Remote Sensing, Karlsruhe, Germany
[3]European Organisation for the Exploitation of Meteorological Satellites (EUMETSAT), Darmstadt, Germany

We would like to thank the reviewer for the valuable and very constructive comments and the effort that was put into reviewing the manuscript. We think the points raised by the reviewer helped improving the main message of the manuscript, substantially. Please find the point-to-point reply below.

5   This paper examines two-slightly different regional cloud mask algorithms using the high-resolution broadband visible channel from MSG SEVIRI instrument for a region in Paris and its vicinity. Both algorithms started with pixel-level and solar zenith angle binned histogram of reflectance. The localized algorithm (LECDA) uses a localized (down to pixel-level) threshold based on a gaussian fitted reflectance histogram, while the regional algorithm (RECDA) uses the maximum of LECDA thresholds for the entire domain. The RECDA algorithm is considered as independent of surface albedo.

10   The paper claims slightly better performance of RECDA method compared with measurements of a ground Cloudnet station. In addition, the cloud fraction from RECDA algorithm is shown to be able to demonstrate the impact of urban heat effect on fog formation from the city of Paris.

  The study is interesting as high-resolution cloud detection algorithm will enable the study of impact on cloud formation due to various natural and anthropogenic factors in very small scales. However, I am not convinced that RECDA is a better algorithm

15 than the LECDA algorithm due to the following reasons:

1. Both algorithms have pros and cons as demonstrated by better POD, larger FAR in LECDA and poor POD, lower FAR in RECDA since the threshold in RECDA is mostly higher than those used in LECDA. The large contrast in POD and FAR of the two algorithms and relatively insignificant difference in overall scores (PC, CSI, HSS) indicate the more

20     fundamental difference of the two algorithm lies in the choice of more clear-conservative or cloud-conservative rather than whether fine tuning of local threshold is better or worse. Therefore what is more important in this case depends on the application. Does the application require to have high POD or low FAR or an overall better score?

    We agree with the reviewer that the performance measures calculated in this study should not be used to show that one approach is better than the other. As pointed out by the reviewer for this approach we do not require to perform better in

terms of POD or FAR than LECDA. Indeed, we would expect LECDA to perform better in a local validation (it is locally tuned after all). The higher suitability of RECDA compared to LECDA for this application is shown in Figure 4 of the manuscript where the difference of both approaches shows a clear signal over forest areas. We think that the dependence of LECDA on the surface signal is better conveyed in Figure 1 (Figure 5a in the manuscript) where the difference of LECDA and RECDA is plotted as a function of clear sky surface reflectance. 75 % of the variability of the difference between RECDA and LECDA can be explained by the surface reflectance. It is assumed that the dependence on the surface reflectance is mostly attributed to LECDA, while a small dependence of RECDA on the surface reflectance may occur in conditions of thin liquid clouds. Thus, RECDA being more robust than LECDA is better suited for this kind of application.

The validation of both approaches with ground truth using the traditional measures and scores is intended to provide an additional context where the developed regional approach can be compared to the local one. The manuscript was modified as follows:

"The general dependence of LECDA on the surface reflectance is shown in Figure 5a) where the difference of LECDA and RECDA is a function of clear sky surface reflectance. 75 % of the variability of the difference between RECDA and LECDA can be explained by the surface reflectance. It is assumed that the dependence on the surface reflectance is mostly attributed to LECDA, while a small portion may be attributed to RECDA in conditions of thin liquid clouds. Thus, RECDA being more robust than LECDA is better suited for this kind of application assuming the presence of thick clouds."

"The validation of the two cloud masks $CM_{loc}$ and $CM_{reg}$ with respect to the Cloudnet data seeks to compare both approaches with ground-truth data using conventional statistics. However, it is not intended to prove the better applicability of one over another for the analysis of land surface effects on boundary layer clouds as this was shown in Section 4, e.g. Figure 4 and 5."

"It is notable that a main difference in validating both approaches originates from the definition of a more cloud-conservative threshold vs. a more clear-conservative threshold."

2. The LECDA tries to follow the distribution of clear sky pixels, as the algorithm is derived from clear sky portion of the GMM, while RECDA tries to preserve the cloud distribution as it assumes the cloudy portion of Gaussian distribution is fixed. This assumption will be more appropriate for thick clouds but not thin clouds as surface reflectance could also impact cloudy sky reflectance.

The calculation of LECDA and RECDA is effectively based on the clear sky distribution while the cloud distribution is rather an assumption. We agree with the reviewer that thin liquid clouds - ice clouds are mostly excluded in the preprocessing - can introduce a small bias depending on the surface signal that should be minor compared to the influence of the surface signal in LECDA. As we cannot exclude an effect due to thin liquid clouds, subpixel clouds or cloud edges

[Figure]

**Figure 1.** a) Linear regression: Difference of cloud fraction anomalies (LECDA-RECDA) vs. clear sky reflectance (average of SZA bins 67-77). Bright colors represent a higher probability density of the data points using Gaussian kernels.

we have added this limitation now in Section 4.3. of the manuscript.

3. The evaluation is only conducted over one location even though the reflectance of the selected location is close to the domain mean. As mentioned by the author, the comparison over bright surface would be similar but over dark surface, LECDA is expected to have higher POD and little influence on FAR. Therefore, over the entire domain, it is yet to be seen which algorithm performs better. It may help to compare the cloud mask with other multiple-channel satellite cloud mask products such as MODIS with 1km resolution with full spatial coverage.

With this study we do not intend stating that the performance of RECDA is better than LECDA in terms of POD. We want to show the improved applicability of LECDA compared to RECDA for the analysis of land surface effects on boundary layer clouds. We think that validating both approaches at one location is sufficient to assess and compare both approaches to the ground truth (as done e.g. Roebeling et al. 2008). A validation with an additional satellite-based product would not support the message of the manuscript. The manuscript was modified to clarify this aspect.

"The validation of the two cloud masks $CM_{loc}$ and $CM_{reg}$ with respect to the Cloudnet data seeks to compare both approaches with ground-truth data using conventional statistics. However, it is not intended to prove the better applicability of one over another for the analysis of land surface effects on boundary layer clouds as this was shown in Section 4, e.g. Figure 4 and 5."

4. The Cloudnet cloud sample selection requires 90% of cloud fraction which is very cloud conservative while SEVIRI cloud masks only require 1/9 fraction to be cloud. This mismatch in spatial/temporal cloud fraction could contribute to slightly better performance in relatively underestimated RECDA algorithm as cloud detection rate would be even lower if more partial Cloudnet pixels are selected as cloudy.

This is an important point that we have discussed internally performing multiple threshold tests. Decreasing the cloud net cloud fraction or increasing the SEVIRI cloud fraction required to be classified as cloud will decrease POD together with FAR for both algorithms, RECDA and LECDA. Less clouds will be detected in the cloud mask and less false alarms will occur. Finding the optimum aggregation scale is complicated as we are comparing a vertically and temporally highly resolved point measurements of Cloudnet with a 15 minute SEVIRI snapshot of the cloud top over a larger spatial area. Following the recommendations of reviewer 2 and 1 we now decided using more moderate and objective thresholds. We recalculated the validation measures for both algorithms using a Cloudnet cloud fraction of 50% over a time window of 1 hour (Deneke et al. 2009) and a SEVIRI cloud fraction of 4/9. The results show a degradation of the performances of both RECDA and LECDA with respect to POD, while FAR is reduced. We have modified the manuscript accordingly.

5. The relative performance of RECDA and LECDA might change with the domain size and dominant cloud types in the region. It is well known that a globally fixed threshold does work well. What is the domain size and surface uniformity requirement for the RECDA to perform better than LECDA?

Thanks for raising this point. We expect that the performance of RECDA could be affected by the availability of data (more data per SZA bin), the distribution of dark vs. bright pixel and the variability of the satellite viewing geometry. A profound analysis would be required that is dedicated to test the different domain sizes, regions as well as cloud types. Based on this study we can recommend RECDA for the proposed application and suggested domain size with a clear sky reflectance between 8 and 10 % (see Figure 2.). RECDA will not provide reliable results over regions with clear sky reflectances varying between 8 and 50% as e.g. over agricultural and desert regions. This is now added to Section 4.3.

**Minor comments:**

What is the bandwidth of the HRV of SERVIRI? I saw some website mentions 0.4-09.um instead of 0.4-1.1$\mu m$.

The bandwith of HRV is 0.4 to 1.1$\mu m$ according to Schmetz et al. 2002

What is x in Equation 2)?

This was meant to be a multiplication sign. We deleted it according to the equation guidelines.

P7L185-190. I don't see how RECDA would not create surface albedo dependent bias unless the algorithm only focused on thick clouds (in that case surface albedo doesn't matter). It is a cloud-conservative approach and assumes that reflectance distribution of cloudy pixels does not change. However, due to overlap of clear sky and cloud sky histogram, fixing threshold for cloud distribution (even assuming it does not change) inevitably affect cutoff of clear sky distribution.

We agree with the reviewer that the RECDA approach may suffer from a surface dependent bias in cases where thin liquid clouds are present. We expect that this could introduce a small bias that should be minor compared to the influence of the surface signal in LECDA. We have added this limitation now in Section 4.3. of the manuscript.

Figure 6. How is the anomaly computed? Anomaly with respect to domain averaged mean for all samples or anomaly of individual pixels in fog-prone conditions versus all conditions? Could you plot the same anomaly figure (Fig.6a) from the LECDA method?

We computed the anomaly of individual pixels with respect to domain averaged means constrained by fog conditions. A description was added to the caption of Fig. 6 in the manuscript. The cloud fraction anomaly from the LECDA method shows a comparable pattern to the cloud fraction anomaly from the RECDA method (Fig. 2). The difference of both (Fig. 3) shows a similar pattern as in Fig. 4a of the manuscript.

[Figure]

**Figure 2.** Cloud fraction anomaly from RECDA (a) and LECDA (b) constrained by the following meteorological conditions: low wind speed (<3 ms$^{-1}$), low blh (<300 m), high msl (>1020 hPa).

[Figure]

**Figure 3.** Difference of cloud fraction anomalies (CFloc anomaly-CFreg anomaly) constrained by meteorological conditions as in Fig. 2.

**References**

Deneke, H. M., Knap, W. H., and Simmer, C.: Multiresolution analysis of the temporal variance and correlation of transmittance and reflectance of an atmospheric column, Journal of Geophysical Research Atmospheres, 114, https://doi.org/10.1029/2008JD011680, 2009

Roebeling, R. A., Deneke, H. M.,  Feijt, A. J.: Validation of cloud liquid water path retrievals from SEVIRI using one year of CloudNET observations. Journal of Applied Meteorology and Climatology, 47(1), 206–222. https://doi.org/10.1175/2007JAMC1661.1, 2008

10  Schmetz, J., Pili, P., Tjemkes, S., Just, D., Kerkmann, J., Rota, S., and Ratier, A.: An Introduction to METEOSAT Second Generation (MSG), Bulletin of the American Meteorological Society, pp. 977–992, https://doi.org/10.1175/1520-0477(2002)083<0977, 2002.